# Synergistic Flame Retardant Properties of Polyoxymethylene with Surface Modified Intumescent Flame Retardant and Calcium Carbonate

**DOI:** 10.3390/polym15030537

**Published:** 2023-01-20

**Authors:** Zheng Yang, Xueting Chen, Shike Lu, Zhenhua Wang, Jiantong Li, Baoying Liu, Xiaomin Fang, Tao Ding, Yuanqing Xu

**Affiliations:** 1Henan Engineering Research Center of Functional Materials and Catalytic Reaction, Kaifeng 475004, China; 2College of Chemistry and Chemical Engineering, Henan University, Kaifeng 475004, China

**Keywords:** modified ammonium polyphosphate, synergistic flame retardancy, polyoxymethylene, titanate coupling agent, calcium carbonate

## Abstract

Ammonium polyphosphate (APP) was successfully modified by a titanate coupling agent which was compounded with benzoxazine (BOZ) and melamine (ME) to become a new type of intumescent flame retardant (Ti-IFR). Ti-IFR and CaCO_3_ as synergists were utilized to modify polyoxymethylene (POM), and the flame-retardant properties and mechanism of the composites were analyzed by vertical combustion (UL-94), limiting oxygen index (LOI), TG-IR, and cone calorimeter (Cone), etc. The results show that Ti-IFR can enhance the gas phase flame retardant effect, while CaCO_3_ further strengthens the barrier effect in the condensed phase. When they were used together, they can exert their performance, respectively, at the same time showing excellent synergistic effect. The FR-POM composite with 29% Ti-IFR and 1% CaCO_3_ can pass the UL-94 V0 level. The LOI reaches 58.2%, the average heat release (Av HRR) is reduced by 81.1% and the total heat release (THR) is decreased by 35.3%.

## 1. Introduction

Polyoxymethylene (POM) is an engineering plastic with excellent physical and mechanical properties and is known as a metal plastic [1]. It is widely used in the electrical and electronic, automotive and construction industries in contemporary society to replace copper, aluminum, zinc, and other metal materials [2]. With the development of modern industry, the demand for POM in the market has soared. Since the beginning of the 21st century, the consumption of POM in the world has reached 540,000 tons with an average annual growth rate of 5–6% [3]. However, with the increase in the demand for POM, its flammable defect is gradually exposed. Once ignited, POM burns violently, and the burning process is accompanied by a large number of burning molten droplets and toxic formaldehyde gas. This is extremely detrimental to its development in the construction and automotive industries [4]. Therefore, the flame-retardant research of POM came into being.

The processing temperature of POM is extremely narrow at 170–200 °C, and it is very easy to decompose at above 185 °C, and the decomposition products will further accelerate it resulting in “unzipper-like” degradation [5]. In addition, POM is very sensitive to acids and alkalis which may also cause its unzipper decomposition [6]. Moreover, the high crystallinity of POM results in its poor compatibility with flame retardant additives leading to an unavoidable and substantial reduction in mechanical properties [7,8]. Therefore, it is quite difficult to obtain ideal POM flame retardants.

Brominated flame retardants (BFRs) are high-efficiency, easy to use, and have good compatibility with polymeric materials which have been used for a long time. However, the harm of BFRs to the environment and human beings has been gradually proposed with the improvement of people’s awareness of environmental protection in recent years, and thus they have been gradually eliminated by the market [5,9,10]. Efficient and environment-friendly flame retardants are the trend of future development. Phosphorus-based [11], nitrogen-based [12], inorganic [13], and organic [14] halogen-free flame retardants have been produced successively. Organic flame retardants have good flame retardancy and good compatibility with polymeric materials. However, the synthesis process is complicated [15]. Inorganic flame retardants such as magnesium hydroxide and aluminum hydroxide, are easy to obtain and very economical. However, the additional amount to meet the demand is large, and the compatibility with polymer is not good, which is easy to cause stress concentration, thereby reducing the mechanical properties [16]. Intumescent flame retardants (IFRs) are regarded as green and environment-friendly ones with the advantages of being halogen-free, low smoke, low toxicity, and no corrosive gas generation. They usually consist of dehydrating agents, char-forming agents, and blowing agents which act synergistically and make it more advantageous than the use of phosphorus-based and nitrogen-based flame retardants alone [17,18]. Phosphorus-containing flame retardants are often used as dehydrating agents, and nitrogen-based flame retardants are often used as foaming agents, so they are also called phosphorus-nitrogen IFRs. They are considered as the most promising flame retardant for POM, but the efficiency is not high enough, and the additional amount sometimes even exceeds the content of POM itself [19]. Lu et al. made ME/RP composite flame retardant by simply mixing red phosphorus and melamine by mechanical grinding, which achieved excellent flame-retardant performance in polyurethane foam and had more advantages than using ME and RA alone [20]. Adding synergists is the easiest way to improve flame retardant efficiency by promoting the synergistic effect between the IFR components [21]. Commonly used flame-retardant synergists mainly include organosilicon compounds [22], red phosphorus and phosphorus compounds [23], zinc borate [24], metal oxides [25], layered nanomaterials [26], etc.

Calcium carbonate (CaCO_3_) is a common chemical substance on the earth, which is an inorganic salt mineral and has a certain flame-retardant effect. It has a wide range of sources and is a very economical flame-retardant synergist [27]. In addition, CaCO_3_ can still improve the processability and heat resistance of the polymer to a certain extent [28]. Sarang et al. [29]. studied the chemical action mode of calcium carbonate nanoparticles combined with ammonium polyphosphate as a flame retardant in polypropylene. Due to the interaction of additives, which impedes the advancing flame, the combustion rate is reduced, and the fire resistance is greatly improved. However, as an inorganic flame-retardant synergist, CaCO_3_ has poor compatibility and dispersity in polymeric materials [30]. Surface modification technology can be used to make up for this deficiency on the premise of maintaining the original properties of products [31]. There are mainly surface chemical reaction method, surface grafting method, surface composite method, and other technologies [32]. Coupling agents are the most commonly used surface modifiers, such as titanate coupling agents, silane coupling agents, and aluminate coupling agents, etc. [33]. The element of Ti, Si, and Al in the molecule make them have potential flame retardancy [34].

In our previous study, ammonium polyphosphate (APP), benzoxazine (BOZ), and melamine (ME) were found a highly effective IFR for POM [35]. In order to further enhance flame retardant efficiency and the compatibility of additives with the POM matrix, titanate coupling agent 201 was used to modify the surface of APP and CaCO_3_ as synergists in this paper. The effects of surface-modified APP and CaCO_3_ on the flame-retardant properties of POM were explored, and the synergistic flame-retardant mechanism of CaCO_3_ and IFR was studied.

## 2. Experimental Section

### 2.1. Materials

POM(M90) and Antioxidant (Model 1010) was provided by Kaifeng Longyu Chemical Co., Ltd. APP (TF-201, crystal form II, n ≥ 1000) was purchased from Shifang Taifeng New Flame Retardant Co., Ltd., and its SEM image is shown in (Figure 1a). BOZ was synthesized in our laboratory, and its chemical structure is shown in (Figure 2) [35]. ME is provided by Henan Zhongyuan Dahua Group Co., Ltd.China. CaCO_3_ (Common type, 3000 mesh) was purchased from Shandong Jinrunze New Material Technology Co., Ltd., Shandong, China and its SEM image is shown in (Figure 1b). Titanate coupling agent (PN-201) was purchased from Nanjing Pinning Coupling Agent Co., Ltd., Nanjing, China.

### 2.2. Preparation of Ti-APP and Ti-CaCO_3_

Pour APP (200 g) into a three-neck flask, add industrial alcohol (400 mL), and stir at 50 °C water bath for 1 h to disperse APP to a uniform suspension. Then, 10 mL ethanol-water mixed solvent was prepared in equal proportion. PN-201 (2 g) was added into the mixed solvent and dispersed by ultrasonic wave for 0.5 h. Finally, the dispersed PN-201 was slowly dropped into the APP suspension and continued to stir at 50 °C for 1 h. After the reaction, the APP suspension was filtered by negative pressure and washed with anhydrous ethanol for 2–3 times. Finally, put the cake in the air blast oven and dry it for 12 h at 80 °C, then modified APP was obtained, namely Ti-APP.

The modified CaCO_3_ (Ti-CaCO_3_) was prepared with 20 g CaCO_3_, 100 mL industrial alcohol, and 2 g PN-201 according to the above program of modified APP.

The mechanism of modified APP and CaCO_3_ is shown in (Figure 3), and the preparation process is shown in (Figure 4).

### 2.3. Preparation of FR-POM Composites

The IFR consists of APP, BOZ, and ME in the mass ratio of 10:2:3 in line with our previous optimization results. Here, Ti-APP is compounded with BOZ and ME at the same mass ratio to form a new intumescent flame retardant named as Ti-IFR instead of IFR. Additionally, the total amount of additives was kept at 30 wt%.

According to the formula in Table 1, POM and a certain proportion of the flame-retardant additives are uniformly mixed in a high-speed mixer and then put into an extruder for melt blending and granulation. The melting temperature of POM was maintained at 165–175 °C. The pellets were placed in a blast drying oven and dried at 80 °C until moisture-free. Finally, the fully dried pellets were put into the injection molding machine for standard test specimens.

### 2.4. Characterization

Fourier transform infrared spectroscopy (FTIR) analysis was performed by a VERTEX 70 spectrophotometer (Bruker Spectrometer, Germany). Potassium bromide tablets were used with an optical range of 400–4000 cm^−1^ and a resolution of 4 cm^−1^.

X-ray powder diffraction (XRD) analysis was performed by Bruker D8 Advance. The 2θ angle range was 5–50°, and the scan time was 15 min.

LOI data of all samples were obtained in accordance with GB/T 2406.2 standard on a smart oxygen index fume hood integrated machine (Suzhou Testech Testing Instrument Technology Co., Ltd., Suzhou, China.), and the sample size was 80 mm × 10 mm × 4 mm.

The UL-94 test was performed in accordance with the GB/T 2408-2008 standard on a horizontal and vertical combustion tester (Testech (Suzhou)) Testing Instrument Technology Co., Ltd., Suzhou, China.) using a sample with a geometric size of 125 mm × 12.5 mm × 3.2 mm and 125 mm × 12.5 mm × 1.6 mm. Test results were classified as Class V-0, V-1, V-2, or No Class (NR).

Thermal degradation-infrared analysis of the samples was carried out in a combined thermogravimetric analyzer (TGA/SDTA 851, Mettler-Toledo) and Fourier infrared tester (INVENIOS, Bruker), heated from room temperature to 800 °C under a nitrogen and air atmosphere. The weight of the test sample was kept at about 5 mg, the gas flow rate was 50 mL/min, and the heating rate was 10 °C/min.

Cone calorimetry (Fire Testing Technology Ltd., East Grinstead, West Sussex, UK) using ISO 5660 standard to study the combustion behavior at an incident radiant flux of 50 kW/m^2^. The dimensions of the samples are 100 mm × 100 mm × 6 mm.

Surface topography of the coke slag measured by cone calorimetry was analyzed by scanning electron microscopy (SEM, JEOL JSM-7610F). The test voltage is 5 kV.

The degree of graphitization of the carbon layer was tested by a laser microscope Raman spectrometer (Renishaw inVia, Renishaw, London, UK), 532 nm excitation wavelength, spectral range: 1000–2000 cm^−1^

## 3. Results and Discussion

### 3.1. Characterization of Ti-APP and Ti-CaCO_3_

Thermogravimetric analysis characterizes the mass change in the material during the temperature-programmed process. It reflects the sensitivity and stability of the material to heat [36]. Materials with higher initial decomposition temperature and carbon residue rate tend to have better flame-retardant properties. (Figure 5) and (Table 2) are the thermogravimetric test curves and characteristic parameter tables of modified APP and modified CaCO_3_, respectively. It can be found from Figure 4 that the modified APP and CaCO_3_ are significantly different compared with the unmodified ones. The characteristic data in Table 2 can more intuitively show the difference before and after modification. The modified APP (Ti-APP) still decomposes twice during the temperature programmed process, which is consistent with the decomposition process of unmodified APP. However, the heat resistance of Ti-APP was improved after modification, and its initial decomposition temperature (*T*_-5%_, *T*_-10%_) and the first fastest decomposition temperature (*T_max_*) were increased by about 10 °C, respectively. The fastest decomposition temperature for the second time even increased by 25 °C. Unfortunately, the residual carbon rate of Ti-APP is lower than that of APP.

After modification, Ti-CaCO_3_ turned to decompose in two steps similar to APP during the temperature programmed process of 25–800 °C, and its heat resistance also changed obviously. The initial decomposition temperature (*T*_-10%_) was advanced by 21 °C which may be caused by the decomposition of the titanate coupling agent coated on its surface, but the fastest decomposition temperature (*T_max_*) in the second step was delayed by 21 °C indicating that the decomposition process of Ti-CaCO_3_ is more gradual, and the addition of titanate slows down its decomposition process.

The infrared spectra of APP, CaCO_3_, and their modification are shown in (Figure 6). PN-201 mainly has two characteristic peaks. From 1030 cm^−1^ to 740 cm^−1^, the infrared absorption decreases continuously and shows strong absorption corresponding to Ti-O. Additionally, the antisymmetric vibration peak of C-H appears around 2934 cm^−1^ [37]. In Figure 5, Ti-APP exhibits the characteristic absorption peak of Ti-O at 1015 cm^−1^, while Ti-CaCO_3_ shows it at 1064 cm^−1^. The absorption of Ti-CaCO_3_ at 2929 cm^−1^ is attributed to the C-H antisymmetric vibration peak. Comparatively speaking, the C-H absorption of Ti-APP appears at 3054 cm^−1^ showing a blue shift, which may be due to the shift of the absorption peak to the high wave number caused by the induction effect. The characteristic peak of PN-201 was detected in the modified APP and CaCO_3_, indicating that it was successfully loaded on APP and CaCO_3_.

The XRD spectra of APP, CaCO_3_, and their modification are shown in (Figure 7). The characteristic diffraction peaks of CaCO3 and Ti-CaCO3 were found at 23°, 29°, 36°, 39°, 43°, 47.5°, 48.5° [38]. Both APP and Ti-APP found their characteristic diffraction peaks at 16.7°, 15.5°, 20°, 22°, 26°, 27.6°, 29°, 30.6°, 35.7°, 36.5°, 38° [39]. According to it, the spectra of Ti-APP and Ti-CaCO_3_ after modification have no obvious change from those before modification. This indicates that the modification of APP and CaCO3 by PN-201 does not change their crystal morphology and has no obvious effect on their structure and state.

### 3.2. Flammability of FR-POM Composites

UL-94 and LOI tests are often used to initially characterize the combustion properties of materials. According to the analysis of them, the char formation of the intumescent material during the combustion process, and the compactness of the char formation can be identified [40]. The test results of UL-94 and LOI are shown in (Table 3).

Pure POM has virtually no flame retardancy. Once ignited, it will burn violently and produce combustion dripping, so it fails to reach the flame-retardant level and its LOI value only reaches 15%. After IFR is added, the combustion drip–drop phenomenon disappears, and the LOI of P1 reaches 46.6%, which is more than three times that of P0, indicating that P1 has good carbon formation performance in the combustion process. However, UL-94 results showed that P1’s second ignition time was too long to reach flame retardant level. It indicates that the carbon layer cannot effectively block the release of combustible gas inside the material, and the density is still insufficient. After APP was modified by PN-201, the flame-retardant performance was further improved. The P2 achieves a V1 rating with an improved LOI value of 51%. The synergistic effect of adding CaCO_3_ to the IFR makes P3 reach the UL-94 V1 level, and the LOI value is increased to 53.6%. It is shown that whether the modified APP (Ti-APP) or the synergistic effect of CaCO_3_ is used, the carbon formation of FR-POM composites can be further enhanced, and the compactness of the carbon layer can be significantly improved.

IFR with Ti-APP as acid source (Ti-IFR) synergistic CaCO_3_ achieved the most excellent synergistic flame-retardant effect among all the FR-POM. The P4 achieves a UL-94 V0 rating and a high LOI value of 58.2%, far better than P2, P3, and even P5. The performance of P5 with modify CaCO_3_ (Ti-CaCO_3_) and Ti-IFR is only inferior to that of P4. It also reaches the UL-94 V0 level, but its LOI of 56.8% is slightly lower than P4. It may be due to the decrease in thermal stability of Ti-CaCO_3_ leading to a slight decrease in carbon formation and carbon layer compactness of P5, compared with unmodified CaCO_3_.

### 3.3. Thermogravimetric Analysis

The TG and DTG curves of FR-POM composites under N_2_ and air atmospheres are shown in (Figure 8), and the thermogravimetric data are summarized in (Table 4 and Table 5). It can be seen from Figure 7 that the decomposition trends of FR-POM composites under N_2_ and air atmospheres are almost the same, both showing one-step decomposition. However, the N_2_ atmosphere seems to be more conducive to the formation of carbon residues, which may be due to the participation of oxygen in the air atmosphere to generate CO_2_, thereby reducing the amount of carbon residues [41]. In both atmospheres, the initial decomposition temperature *T*_-5%_ of P0 is around 233 °C, *T*_-10%_ is around 248 °C, and the fastest decomposition temperature is about 295 °C. Moreover, the whole decomposition process of P0 was smoother than others, the decomposition was more thorough, and there was basically no carbon residue left. After adding different flame retardants, the thermal decomposition of FR-POM is very similar. The initial decomposition temperature *T_-5%_* is increased by about 20 °C compared with P0, but *T_max_* is only about 260 °C, which is very close to *T*_-10%_ and *T*_-5%_. The thermal resistance of P1, P2, P3, P4, and P5 is improved, and the thermal decomposition temperature range is narrow. It may be that the addition of flame retardants not only improves the heat resistance of composite materials but also disturbs the original structured molecular structure of POM so that its decomposition process is very rapid and shows a narrow decomposition range.

The carbon residue rate at 800 °C reflects the carbon-forming ability of the materials. In the N_2_ atmosphere, the addition of IFR makes the carbon residue rate of P1 reach 15.2%. After using Ti-IFR, the carbon residue rate of P2 is further improved, reaching 17.1%. While the carbon residue rate of P3 with CaCO_3_ as a synergist is only 14.0% indicating that Ti-APP can improve the carbon residue rate more efficiently than CaCO_3_. Unexpectedly, the synergistic system of CaCO_3_ and Ti-IFR achieved the best carbon formation effect. The carbon residue rate of P4 reached 18.2%, which was 19.7% higher than that of P1. However, by combining Ti-CaCO_3_ with Ti-IFR, the carbon residue rate of P5 is slightly lower than that of P4. It shows that Ti-IFR and CaCO_3_ have an excellent synergistic flame-retardant effect and excellent carbon-forming properties. In addition, the FR-POM composites also showed similar carbon-forming ability in the air atmosphere. Furthermore, the carbon residue rate of P3, P4, and P5 with calcium carbonate in the AIR atmosphere is apparently higher than that of P1 and P2 without it, indicating that CaCO_3_, whether modified or not, is highly efficient to promote the carbon-forming and synergistic effect with IFR under air atmosphere.

### 3.4. Cone Calorimeter Research

Cone calorimetry (CONE) is a method to record information about the combustion state of materials in a simulated real fire environment. As a very reliable test, it has been widely used in the evaluation of flame-retardant properties of composite materials such as various engineering plastics, general plastics, foam materials, and wood materials [38]. Characteristic data such as time to ignition (TTI), heat release rate (HRR), total heat release (THR), effective heat of combustion (EHC), total smoke generation (TSP), and specific extinction area (SEA) can be obtained. Through these data, an objective analysis of the material’s heat release, gas release, and flame-retardant mechanism can be carried out. The curves of HRR, THR, TSP, and mass loss rate (MLR) of FR-POM composites as a function of time are shown in (Figure 9). The characteristic data obtained from the cone calorimetry test are listed in (Table 6).

As shown in Figure 8, P0 was ignited at 43 s, the peak heat release rate (Pk HRR) reached 361.2 kW/m^2^, and the THR amounted to 140.7 MJ/m^2^. The HRR curve peaks in a very short time, so the average heat release rate (Av HRR) is as high as 267.6 kW/m^2^. After the introduction of flame-retardant additives, the ignition time of all FR-POM was significantly earlier than that of P0, probably due to the decomposition of IFR at high temperatures to perform its flame-retardant effect and lead to the earlier decomposition time of FR-POM composites.

The flame-retardant effect of IFR is very effective and significant. The Pk HRR value of P1 is only 124.6 kW/m^2^, and the THR value is only 107.9 kW/m^2^. Compared with P0, the Pk HRR and THR of P1 are decreased by 65.5% and 23.3%, respectively. In addition, because of the prolonged burning time, the Av HRR value of P1 decreased by 77.6%. The HRR value characterizes the heat release rate per second of the material and is closely related to the total heat released (THR) during the combustion process. Obviously, the higher the HRR and THR of the material, the higher the fire risk.

After using Ti-IFR instead of general IFR, the Pk HRR and the mean effective heat of combustion (mean EHC) of P2 decreased by 16.3% and 8.1%, respectively, compared with that of P1, but both Av HRR and THR increased slightly. Moreover, the specific extinction area (SEA) increased by 81.4%, the total smoke emission (TSP) of P2 is 2.1 times that of P1, while the amount of carbon residue is 0.8 times of P1, which fully shows that Ti-IFR plays a flame-retardant role more in the gas phase than IFR.

Compared with P1, the Pk HRR of P3 decreased by 19.5%, and its Av HRR and THR also decreased a little. Meanwhile, the mean EHC was basically unchanged, and SEA increased slightly. In addition, the amount of residual carbon of P3 is two times that of P1, but the total smoke emission is almost the same. It is proved that the flame-retardant effect was exerted mainly in the condensed phase. It is the large amount of carbon residue that effectively isolates the contact between the combustible gas inside the matrix, oxygen, and the external heat so that the combustion process tends to stagnate.

P2 and P3 show different flame-retardant mechanisms. The former with Ti-IFR further strengthens the gas phase flame retardant effect of IFR, while the latter with CaCO_3_ further improves the condensed phase flame retardant effect. In order to enhance both gas phase and condensed phase performance at the same time, Ti-IFR was compounded with CaCO_3_ to flame retardant POM. As a result, the Pk HRR, Av HRR and THR values of P4 decreased by 75.8%, 81.1%, and 35.3% compared to pure POM, respectively. The flame-retardant effect is far better than P1, P2, and P3. Its mean EHC was slightly decreased, and the SEA was increased compared with P3, indicating a gas phase flame retardant mechanism. Moreover, compared with P2, the mean EHC of P4 was basically unchanged, the SEA and TSP were significantly reduced, and the amount of carbon residue was increased by 2.6 times, which is an obvious condensed phase flame retardant mechanism. Therefore, P4 achieves an excellent effect by combining two flame retardant mechanisms that enhance simultaneously.

Like P4, P5 has excellent flame retardancy which uses PN-201 modified CaCO_3_ (Ti-CaCO_3_) and Ti-IFR to synergize flame retardant POM. Its Pk HRR, Av HRR, and THR decreased by 73.3%, 75.4%, and 16% compared with pure POM, respectively. However, its flame-retardant properties are slightly lower than P4, although it is better than other samples. The amount of residual carbon was significantly reduced, and the mean EHC was increased indicating that the condensed phase flame retardancy is restrained after loading Ti-CaCO_3_. It may be because of the addition of Ti-CaCO_3_ that the content of PN-201 in the system increased slightly as a whole breaking the original equilibrium ratio of the gas phase and condensed phase, resulting in a decrease in synergy.

### 3.5. Carbon Layer Analysis of FR-POM Composites

In order to further analyze the integrity and strength of the carbon layer, the surface carbon layer of the FR-POM composite after cone calorimetry was analyzed by digital photographs, field emission scanning electron microscopy (SEM) and laser Raman spectroscopy (LRS) [42,43,44]. Digital photos, SEM pictures and LRS spectra are shown in (Figure 10, Figure 11 and Figure 12) respectively.

Composites with IFR generate large amounts of gas during combustion due to the presence of blowing agents. If the carbon layer has good compactness and high strength, it is difficult for all the gases to break through and will inevitably expand. Therefore, the expansion height of the residual carbon can often preliminarily judge the strength and compactness of the carbon layer, as well as the flame-retardant performance [45]. As shown in the digital photo of carbon residue in Figure 10, pure POM has almost no carbon residue. After adding flame retardant filler, the carbon residue increases significantly, especially for the sample added with calcium carbonate. Ti-IFR furthers the flame-retardant performance in the gas phase, so the carbon layer of P2 is a little fragmented with no obvious difference from that of P1. While P3, P4, and P5 added with CaCO_3_ mainly act in the condensed phase, their carbon layers are obviously denser and higher than P1. P4 and P5 have the best flame-retardant properties in CONE due to the synergistic effect which is also reflected in digital photos in terms of carbon layer strength, swelling height, and compactness. 

The digital photos of carbon residues are used to preliminarily judge the strength and compactness of the carbon layer from the macroscopic point of view, while the SEM photos are observed from the microscopic point of view. In the observed field of SEM image as shown in Figure 11, there are many pores in the carbon layer of P1, accompanied by a few cracks. P2 shows a fragmented carbon layer like P1, but basically has no holes and still has a certain degree of compactness. It may be related to its gas phase flame retardant mechanism. The carbon layer of P3 is much denser than that mentioned above, with very few cracks and sporadic holes in the field of view. This is closely related to its condensed phase flame retardant mechanism. In the field of view of P4, there are basically no holes and cracks, and the surface of the carbon layer is smooth and flat, which is consistent with the results of its flame-retardant properties. The carbon layer of P5 has a few holes, but basically no cracks.

The LRS of the carbon layer mainly shows two peak positions. As shown in Figure 12, one is around 1350 cm^−1^ (D peak), which represents lattice defects in the disordered carbon structure. The other is around 1580 cm^−1^ (G peak), representing the sp^2^ hybridization of the ordered carbon structure with in-plane stretching vibrations. The graphitization degree of the carbon layer is usually indicated by the peak area ratio (I_D_/I_G_) of the D peak to the G peak. Obviously, the smaller the I_D_/I_G_ value, the higher the degree of graphitization of the carbon layer, which is more beneficial for the surface carbon layer to resist flame burning [46]. The I_D_/I_G_ values of P1 and P2 are 0.72 and 0.75 with little difference indicating that Ti-IFR has no contribution to the degree of graphitization of the carbon layer. While the I_D_/I_G_ values of P3, P4, and P5 are significantly lower than that of P1 and P2. This demonstrates that CaCO_3_, whether modified or not, can improve the degree of graphitization of the carbon layer. Especially, P4 has the highest degree of graphitization with the smallest I_D_/I_G_ value of 0.57 among all the samples, which is consistent with its flame-retardant performance.

From the FTIR spectrum of the carbon layer (Figure 13), it can be clearly found that the residual carbon spectra mainly include the absorption peak of OH (3431 cm^−1^), C=O (1630 cm^−1^), P-O (1184 cm^−1^) and P=O symmetric stretching vibration (985 cm^−1^). P0 has obvious -OH and C=O peaks at 3431 cm^−1^ and 1630 cm^−1^. As to the FR-POM samples, their FTIR spectrum is very similar. APP in the additives can produce metaphosphoric acid and pyrophosphoric acid compounds at high temperatures, which catalyzed the ring-opening polymerization of the BOZ carbon agent but also promoted the dehydration and carbonization of the matrix. Thus, significantly weakened the -OH and C=O peaks of P1-P5, and enhanced the absorption peaks of P-O and P=O. After adding CaCO_3_, the peak at around 1184 cm^−1^ of P3–P5 decreased obviously and became a broad one, which is different from that of P1 and P2. It indicated that CaCO_3_ participates in the formation of the carbon layer and changes the mechanism of carbon formation. The reason may be due to the degradation of CaCO_3_ to CaO at high temperatures, which acts as a strong alkaline catalyst to promote the esterification and cross-linking reaction resulting in a high-quality carbon layer.

The EDS data in (Table 7) reveal the main components of the expanded carbon layer. The stable and dense expanded carbon layer is mainly formed by the elements of C, N, O, P, and Ca. The carbon-to-oxygen ratios (C/O) of the carbon layer on the surface of P1-5 are all around 0.5 and greatly lower than their corresponding inner carbon layer. This is because the carbon layer on the surface burns more fully in contact with the flame with a higher oxidation degree to form a more heat-resistant and high-temperature-resistant oxide to protect the inner carbon layer from being burned. The change in the C/O ratio of the inner carbon layer of all samples strongly proves this point. P4 has the best flame-retardant performance, and its surface carbon layer has strong compactness. Therefore, the protection of the inner carbon layer is the best, the C/O of the inner carbon layer is 2.7, and the degree of oxidation is extremely low. In addition, no Ti element was detected in all samples, probably due to the fact that it mainly functions in the gas phase and is used in a very small amount.

### 3.6. Gaseous Products Analysis

The TG-IR combination method was used to obtain information on gas volatiles in FR-POM composite as a function of temperature. The 3D spectra of P0 and P4 are shown in (Figure 14). Since the temperature range of the TG test is 25–800 °C, and the heating rate is 10 °C/min. Therefore, the relationship between time and temperature in Figure 14 is:$$T_{e}=\frac{1}{60}t_{i}\times 10+25$$
(*T*_e_ refers to temperature and t_i_ refers to time).

It can be found from the 3D figure that the types of gas volatiles of P0 and P4 composites are basically the same in the nitrogen atmosphere. Mainly carbonyl compounds (CH_2_O, 1715 cm^−1^, and 1773 cm^−1^), aliphatic lipids (1745 cm^−1^), and hydrocarbon groups such as methyl and ethyl groups (2801–2924 cm^−1^). The results show that FR-POM composites crack mainly in the form of formaldehyde and various small molecular chain alkanes under the condition of high temperatures without oxygen. In addition, the absorption peaks of C-O (1161 cm^−1^), the symmetric stretching vibration of -CH_3_ (1320 cm^−1^), the skeleton stretching vibration of aromatic hydrocarbons (1458–1532 cm^−1^), and the absorption of N-H (3468 cm^−1^) can still be found in the 3D diagram.

In the air atmosphere, P0 and P4 composites have an obvious CO_2_ absorption peak (2358 cm^−1^), which may be due to the participation of oxygen, prompting a part of formaldehyde or small molecular chain alkanes to completely burn, thus producing CO_2_. Moreover, the CO_2_ absorption peak of P4 was significantly lower than that of P0 due to the addition of flame retardant. The results show that the addition of flame-retardant leads to incomplete combustion of the material and locks more C in the carbon layer, thus effectively reducing the corrosion of the gas volatiles. In addition, the decomposition is basically the same as that in the nitrogen atmosphere.

The FTIR spectra of FR-POM composites with the maximum release of gas residues are shown in (Figure 15). It can be found that the decomposition of other groups is basically the same as that of P4. All FR-POM composites showed mainly flammable gas release such as aldehydes, hydrocarbons, and lipids, and no typical titanium-containing substances were found, mainly due to the very small proportion of PN-201 in the overall system.

### 3.7. Flame Retardant Mechanism

Combined with the previous analysis, we put forward the following possible flame-retardant process mechanism. As shown in (Figure 16), when FR-POM composites are continuously exposed to fire or heat sources, the flame-retardant additives begin to take effect mainly in the condensed phase as well as in the gas phase simultaneously.

In the condensed phase, the high temperature makes the surface of the matrix dehydrate and carbonize rapidly, which has a certain protective effect on the interior of the matrix. With the continuous high-temperature erosion, the surface carbon layer will be destroyed, the internal matrix will be affected by the high temperature and the constant release of flammable gas. At this point, the BOZ carbonating agent comes into play. The pyrophosphoric acid and metaphosphate decomposed by APP catalyze the esterification and crosslinking of BOZ, and finally make it become a stable carbon layer with a three-dimensional network structure, which greatly improves the density and strength of the carbon layer, thus blocking the communication channel between the combustible gas in the matrix and the external heat source, and finally interrupts the combustion process [41]. The addition of calcium carbonate further catalyzes BOZ and plays a stronger catalytic carbonization role in the matrix, thus accelerating the formation of the carbon layer.

In the gas phase, the matrix structure is destroyed, and a large number of formats and hydrocarbon-based combustible gases are released, which makes the heat release peak quickly. At the same time, when IFR flame retardant is decomposed by heat, ME quickly decomposes into refractory nitrogen and begins to dilute flammable gas in the gas phase with a large amount of carbon dioxide produced by matrix combustion, thus reducing the intensity of combustion. In addition, TiOxprodued by Ti-APP under high temperature scavenges the high activity free radicals such as H· and HO· bringing the combustion process to a standstill. Thus, a significant flame-retardant performance was achieved (i.e., the gas phase TiO clears the incomplete combustion caused).

## 4. Conclusions

Titanate coupling agent was successfully loaded onto APP. In addition, Ti-APP is compounded with BOZ and ME to become a new type of intumescent flame retardant (Ti-IFR). Compared with IFR, Ti-IFR can enhance the gas phase flame retardant effect. The UL-94 of POM with Ti-IFR is improved from NR level to V1 level, and the LOI is improved from 46.6% to 51.0% compared with that with IFR. Furthermore, the addition of CaCO_3_ into the Ti-IFR system exhibits an excellent synergistic flame-retardant effect. The UL-94 test level was raised to V0, and the LOI was raised to 58.2%. At the same time, the cone calorimetry results show that the Pk HRR, Av HRR, and THR are all significantly reduced. Additionally, Ti-IFR can further its gas phase flame retardant performance, while CaCO_3_ as a synergist promotes the condensed phase flame retardant effect of the system which not only fixed more C in the carbon layer, but also enhanced the quantity of the carbon layer. Therefore, an excellent flame-retardant performance was achieved through the combination of both the condensed phase and the gas phase flame retardant mechanism enhanced simultaneously.

## Figures and Tables

**Figure 1 polymers-15-00537-f001:**
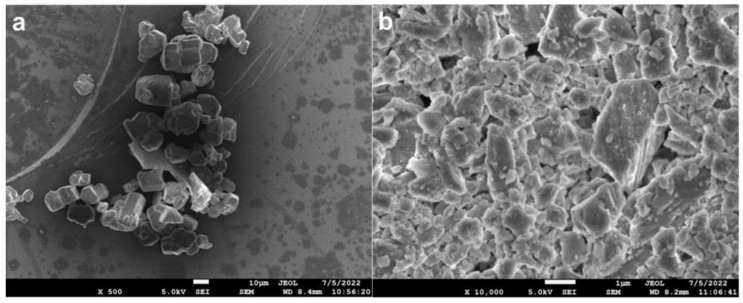
SEM images of commercial APP (**a**) and CaCO_3_ (**b**).

**Figure 2 polymers-15-00537-f002:**
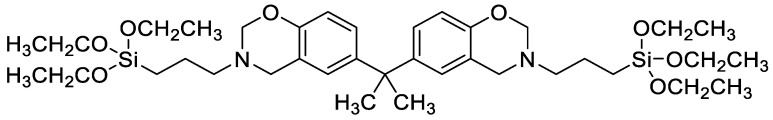
Chemical structure of a novel charring-forming agent (BOZ).

**Figure 3 polymers-15-00537-f003:**
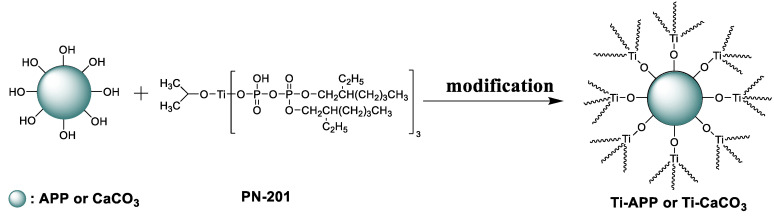
Mechanism of modified APP and CaCO_3_.

**Figure 4 polymers-15-00537-f004:**
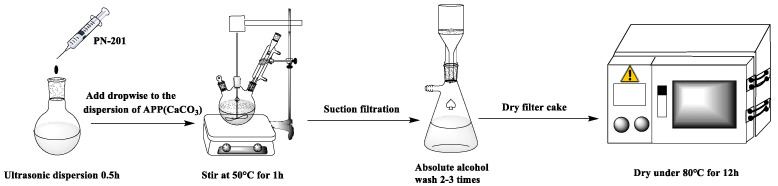
Schematic diagram of the preparation process of modified APP and CaCO_3_.

**Figure 5 polymers-15-00537-f005:**
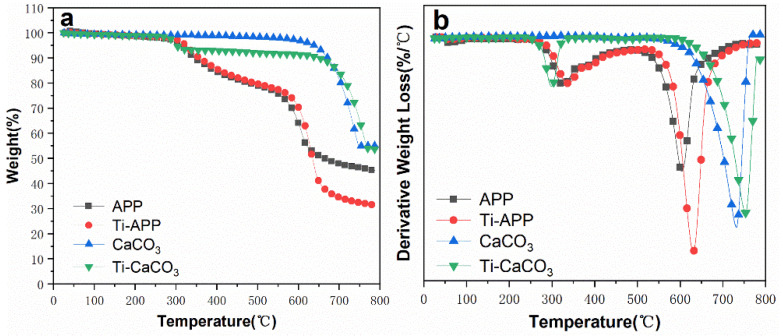
Thermogravimetric analysis of modified APP and CaCO_3_ (**a**) is TG curve and (**b**) is DTG curve).

**Figure 6 polymers-15-00537-f006:**
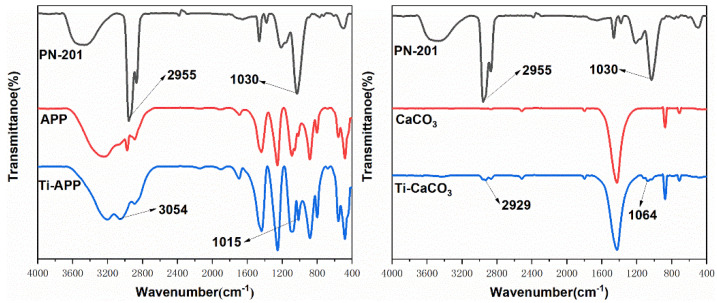
FTIR spectra of APP, CaCO_3_ and their modification.

**Figure 7 polymers-15-00537-f007:**
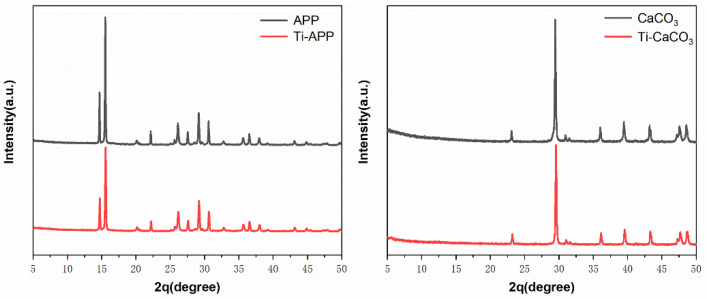
XRD spectra of APP, CaCO_3_ and their modification.

**Figure 8 polymers-15-00537-f008:**
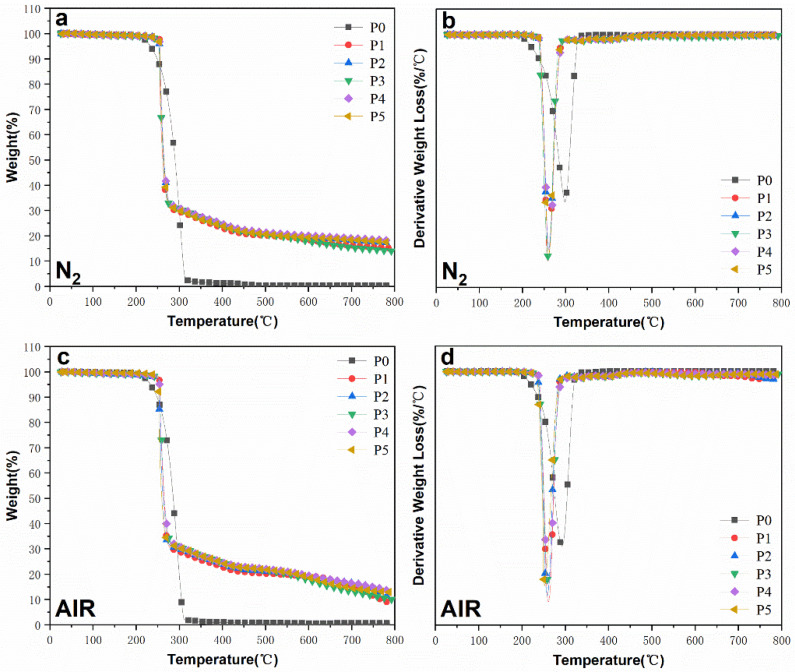
TG and DTG curves of FR-POM composites under N_2_ (**a**,**b**) and AIR (**c**,**d**) atmospheres.

**Figure 9 polymers-15-00537-f009:**
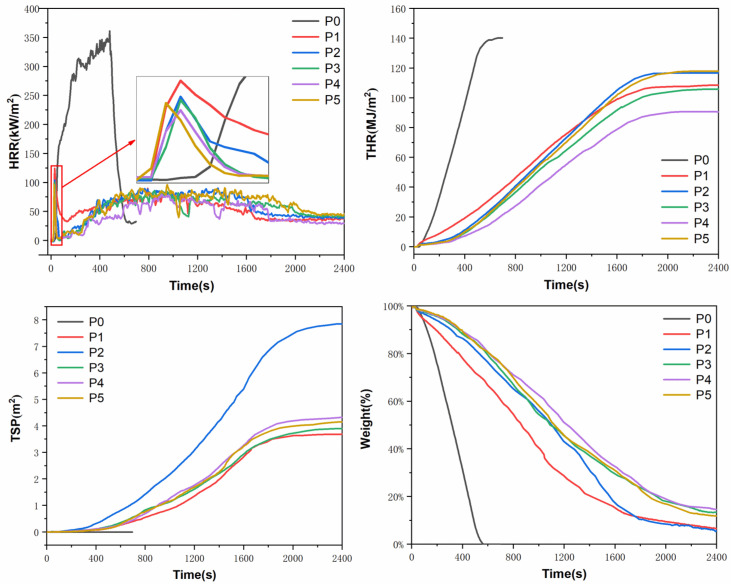
HRR, THR, TSP, and mass loss rate curves of FR-POM composites.

**Figure 10 polymers-15-00537-f010:**
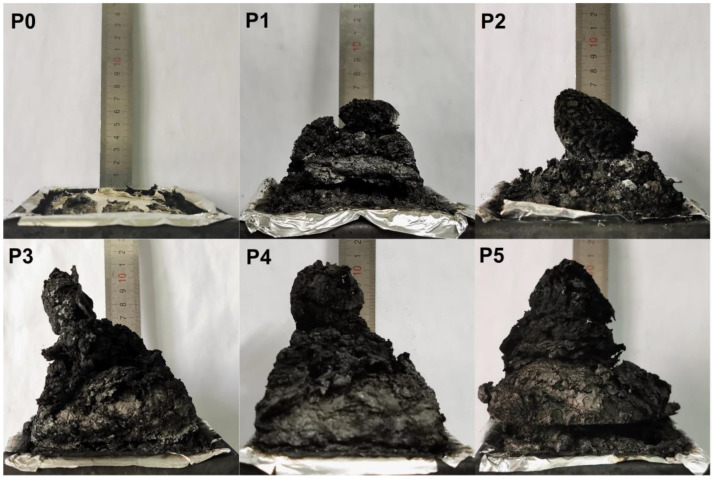
Digital photo of carbon layer after cone calorimetry test.

**Figure 11 polymers-15-00537-f011:**
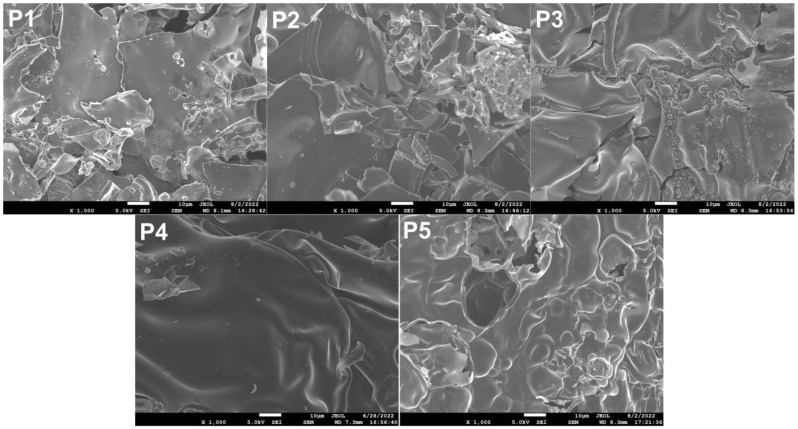
SEM image of surface carbon layer.

**Figure 12 polymers-15-00537-f012:**
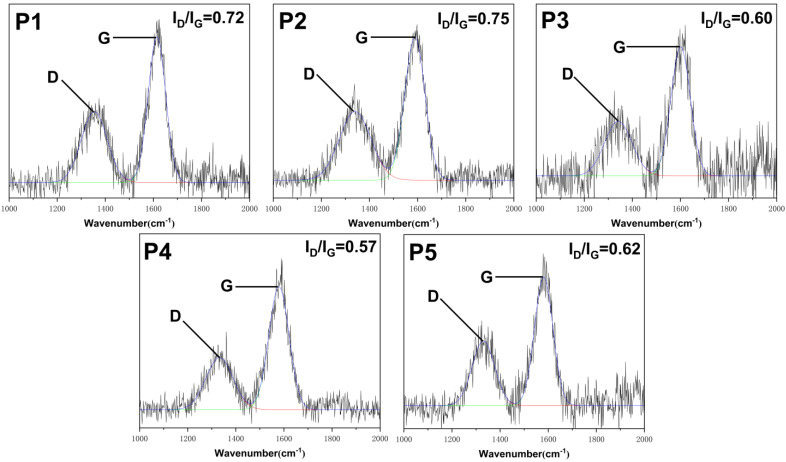
Raman analysis image of surface carbon layer.

**Figure 13 polymers-15-00537-f013:**
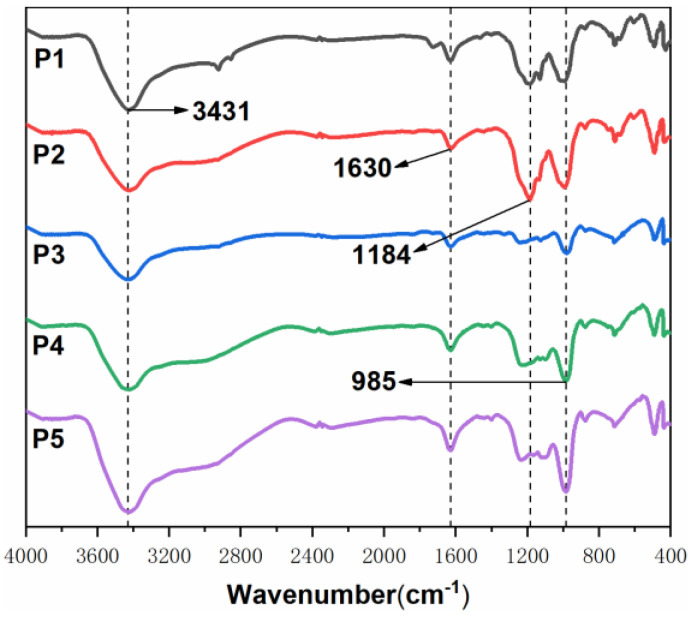
FTIR spectra of residual carbon after cone test.

**Figure 14 polymers-15-00537-f014:**
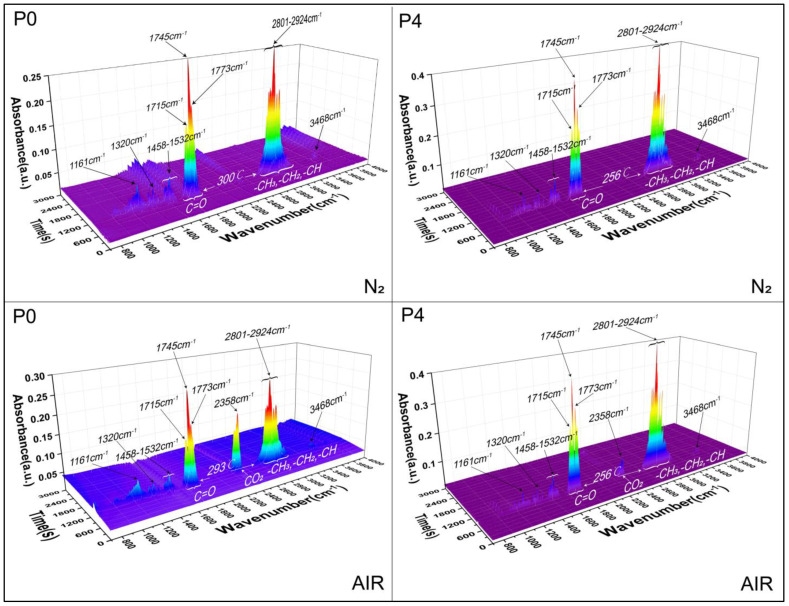
TG-IR 3D diagrams of P0 and P4 composites in N_2_ and AIR atmospheres, respectively.

**Figure 15 polymers-15-00537-f015:**
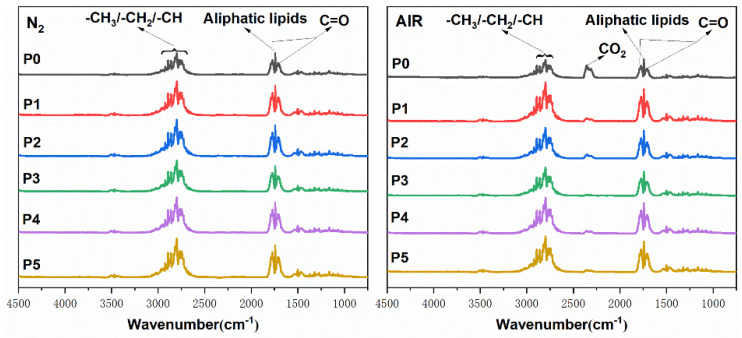
Total decomposition FTIR spectra of FR-POM composites in N_2_ and AIR atmospheres, respectively.

**Figure 16 polymers-15-00537-f016:**
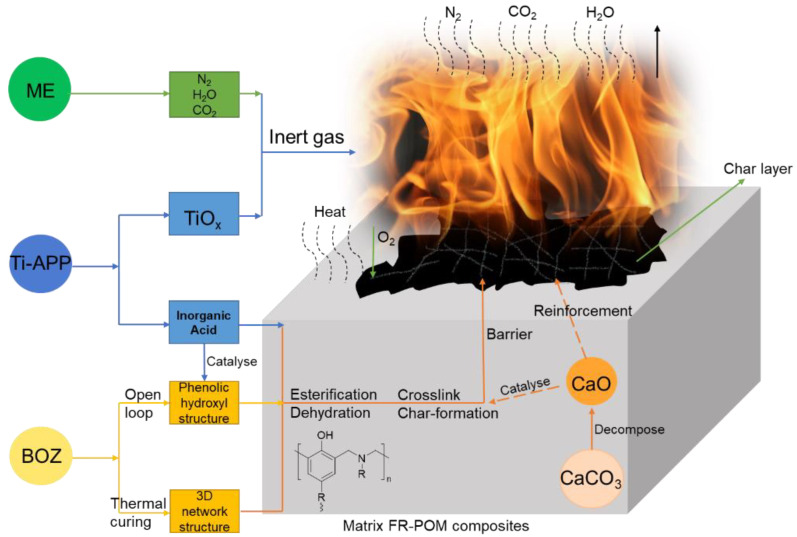
Schematic diagram of flame retardancy mechanism of FR-POM.

**Table 1 polymers-15-00537-t001:** Formulation and nomenclature of flame-retardant composites.

Sample Codes	Composition	POM	IFR (APP)	Ti-IFR (Ti-APP)	CaCO_3_	Ti-CaCO_3_	Antioxidant
P0	POM	99.5	0	0	0	0	0.5
P1	POM/IFR	69.5	30	0	0	0	0.5
P2	POM/Ti-IFR	69.5	0	30	0	0	0.5
P3	POM/IFR/CaCO_3_	69.5	29	0	1	0	0.5
P4	POM/Ti-IFR/CaCO_3_	69.5	0	29	1	0	0.5
P5	POM/Ti-IFR/Ti-CaCO_3_	69.5	0	29	0	1	0.5

**Table 2 polymers-15-00537-t002:** Thermogravimetric data of modified APP and CaCO_3_.

Samples	*T*_-5%_/°C	*T*_-10%_/°C	*T_max_*/°C	Residue (800 °C)/%
APP	313.4	345.4	327.0/604.3 *	45.2
Ti-APP	324.2	357.6	337.5/630.5 *	31.5
CaCO_3_	638.1	671.5	731.5	55.2
Ti-CaCO_3_	301.7	650.3	298.7/752.9 *	53.8

Note: *T*_-5%_ and *T*_-10%_ represent the temperature at 5% and 10% weight loss, respectively; *T*_max_ stands for maximum weight loss rate temperature; “*” represents the maximum weight loss rate temperature during the multi-stage weight loss process of the sample.

**Table 3 polymers-15-00537-t003:** UL-94 and LOI performance data of FR-POM composites.

Samples	Dripping	UL-94			Rating	LOI
		T_1_/s	T_2_/s	T_10_/s		
P0	Yes	-	-	-	NR	15
P1	No	0.3	42.6	214.8	NR	46.6
P2	No	0.3	23.4	118.5	V1	51.0
P3	No	0.2	23.7	119.2	V1	53.6
P4	No	0.3	7.4	38.4	V0	58.2
P5	No	0.6	7.2	39.3	V0	56.8

Note: T_1_ and T_2_ represent the mean values of the first and second ignition times, respectively; T_10_ is the total time of ten fires.

**Table 4 polymers-15-00537-t004:** Thermogravimetric data of FR-POM composites under nitrogen atmosphere.

Samples (N2)	*T*_-5%_/°C	*T*_-10%_/°C	*T_max_*/°C	Residue (800 °C)/%
P0	233.1	248.8	298.0	0.4
P1	254.1	254.2	258.4	15.2
P2	253.6	254.6	260.1	17.1
P3	252.5	253.8	259.7	14.0
P4	254.6	255.6	261.1	18.2
P5	254.4	255.1	260.1	17.5

**Table 5 polymers-15-00537-t005:** Thermogravimetric data of FR-POM composites in AIR atmosphere.

Samples (AIR)	*T*_-5%_/°C	*T*_-10%_/°C	*T_max_*/°C	Residue (800 °C)/%
P0	233.2	248.0	292.4	0.7
P1	254.2	254.9	260.8	8.7
P2	251.5	252.5	258.2	10.7
P3	252.3	254.3	261.2	9.8
P4	254.0	254.7	260.9	13.4
P5	252.2	252.4	259.8	12.8

**Table 6 polymers-15-00537-t006:** Cone calorimetric characteristic data of FR-POM composites.

Samples	P0	P1	P2	P3	P4	P5
TTI(s)	43	25	27	28	27	24
Pk HRR(kW/m^2^)	361.2	124.6	104.3	100.3	87.3	96.4
Av HRR(kW/m^2^)	267.6	60.0	64.5	58.7	50.5	65.7
Pk HRR/TTI(kW/(m^2^·s))	8.4	5.0	3.9	3.6	3.2	4.0
THR(MJ/m^2^)	140.7	107.9	116.3	105.8	91.1	118.2
Mean EHC(MJ/kg)	15.6	14.9	13.7	15.1	14.0	16.1
SEA(m^2^/kg)	1.9	58.2	105.6	63.4	76.1	64.1
Av MLR(g/(m^2^·s))	17.2	4.0	4.7	3.9	3.6	4.1
TSP(m^2^)	0	3.7	7.9	3.9	4.4	4.2
Residue(wt%)	0	6.7	5.5	13.4	14.5	11.9

**Table 7 polymers-15-00537-t007:** EDS data of surface carbon layer and inner carbon layer.

Samples	C/%	N/%	O/%	P/%	Ca/%	Ti/%	C/O
P1	27.7 (44.7)	0 (4.4)	59.9 (43.7)	12.4 (7.2)	0 (0)	0 (0)	0.5 (1.0)
P2	23.7 (54.1)	0 (4.9)	63.0 (34.7)	13.3 (6.3)	0 (0)	0 (0)	0.4 (1.6)
P3	29.2 (51.2)	10.2 (18.3)	52.2 (27.8)	8.1 (2.6)	0.3 (0.1)	0 (0)	0.6 (1.8)
P4	28.7 (61.2)	0 (11.0)	59.6 (22.9)	11.5 (4.8)	0.2 (0.1)	0 (0)	0.5 (2.7)
P5	23.3 (49.5)	0 (11.2)	63.1 (33.2)	12.9 (5.9)	0.7 (0.2)	0 (0)	0.4 (1.5)

Note: A (B) indicates that data A was obtained from the surface of residual carbon, while data B was from the bottom of carbon layer.

## Data Availability

Not applicable.

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
