# Peer review of "Synergistic Flame Retardant Properties of Polyoxymethylene with Surface Modified Intumescent Flame Retardant and Calcium Carbonate"

_polymers, 2023, doi:10.3390/polym15030537_

Round 1
Reviewer 1 Report
The paper could be of interest for the readers of the journal. My comments/suggestions are shown below:
(1) In abstract, there are some problems: tense, combustion performance of pure resin, What guidance can be provided by this work for the future?
(2) CaCO3 is heavy or light or nano?
(3) Line 77, the authors may have accidentally input 41 between word of ‘for’.
(4) Line 83, the authors should provide the reference, which come from themselves.
(5) The FTIR test for CaCO3 is simple, it should further study.
(6) In LOI, TG and Cone tests, it found that the flame retardancy and thermal stability of P4 sample than P5, while CaCO3 and Ti-CaCO3 are selected in P4 and P5 respectively. Why? If authors never give me a reasonable explanation, I think P5 should be deleted. Meanwhile, the title of this work is focused on modified APP.
Author Response
January 10, 2023
Prof. Fang,
Re: Response for manuscript ID: polymers-2144382 “Synergistic Flame Retardant Properties of Polyoxymethylene with Surface Modified Intumescent Flame Retardant and Calcium Carbonate”
Dear reviewer #1,
Thanks for providing us with this great opportunity to submit a revised version of our manuscript. We appreciate the detailed and constructive comments provided by the reviewers. We have carefully revised the manuscript by incorporating all the suggestions by the review panel.
We hope this revised manuscript has addressed your concerns, And look forward to hearing from you.
Sincerely,
The Authors
-------------------------------------------------------------------------------
Reply to Reviewer #1
Dear Reviewers,
Thank you very much for your time involved in reviewing the manuscript and your very encouraging comments on the merits.
Comments:
“The paper could be of interest for the readers of the journal. My comments/suggestions are shown below:
(1) In abstract, there are some problems: tense, combustion performance of pure resin, What guidance can be provided by this work for the future?
(2) CaCO3 is heavy or light or nano?
(3) Line 77, the authors may have accidentally input 41 between word of ‘for’.
(4) Line 83, the authors should provide the reference, which come from themselves.
(5) The FTIR test for CaCO3 is simple, it should further study.
(6) In LOI, TG and Cone tests, it found that the flame retardancy and thermal stability of P4 sample than P5, while CaCO3 and Ti-CaCO3 are selected in P4 and P5 respectively. Why? If authors never give me a reasonable explanation, I think P5 should be deleted. Meanwhile, the title of this work is focused on modified APP”.
We also appreciate your clear and detailed feedback and hope that the explanation has fully addressed all of your concerns. In the remainder of this letter, we discuss each of your comments individually along with our corresponding responses.
To facilitate this discussion, we first retype your comments in italic font and then present our responses to the comments.
Comment 1:
In abstract, there are some problems: tense, combustion performance of pure resin, What guidance can be provided by this work for the future?
Response 1:
Thank you for your advice. As we all know, the flame retardance of POM is a challenging problem. This article is expected to provide new ideas for the construction of high-efficient intumescent flame retardants for POM.
Comment 2:
CaCO3 is heavy or light or nano?
Response 2:
Your suggestion is very necessary. We have added information about the particle size and SEM image of CaCO3 in section of "2.1Materials".
Comment 3:
Line 77, the authors may have accidentally input 41 between word of ‘for’.
Response 3:
Thank you for your careful review of our manuscript. We have corrected the omissions and revised the manuscript again.
Comment 4:
Line 83, the authors should provide the reference, which come from themselves.
Response 4:
Thank you very much. We have inserted the references in the appropriate places.
Comment 5:
The FTIR test for CaCO3 is simple, it should further study.
Response 5:
Thanks for your suggestion, we have elaborated the infrared characterization of CaCO3 in more detail to explain the significance of infrared characterization of it.
Comment 6:
In LOI, TG and Cone tests, it found that the flame retardancy and thermal stability of P4 sample than P5, while CaCO3 and Ti-CaCO3 are selected in P4 and P5 respectively. Why? If authors never give me a reasonable explanation, I think P5 should be deleted. Meanwhile, the title of this work is focused on modified APP”.
Response 6:
Thank you very much. The topic of our manuscript is the modification of APP, but considering that CaCO3 is an inorganic component like APP, and accounts for 1% of the flame retardant system. Therefore, P5 scheme is designed to compare whether the modified CaCO3 is more conducive to the flame retardant system, so as to reflect the optimal combination of Ti-APP+CaCO3.
Sincerely,
The Authors
-----End of Reply to Reviewer #1------

Reviewer 2 Report
Please find suggestion for correction as follow:
|
187 |
Result of FTIR need some explanation about the reason why the functional group that was found to be similar or changed. As well as the newly found spectrum brings what to the sample. |
|
195 |
Similar with XRD. The different diffraction brings what to the sample. Need some explanation and reasons. |
|
213 |
How the UL94 test could determine type of gas released? |
|
217 |
How the compactness property could be related in flammability test? There need to be some explanation about how the different additives added reacted during flammability test & LOI, due to difference in flammability results. |
|
240 |
I think a citation is good to be here. Most of the explanation, normally we add citation as we refer the similarity or the reason of the findings. |
|
248 |
Need some explanation on this. |
|
Overall |
It is a very good and well-established article. Well done. |
Author Response
January 10, 2023
Prof. Fang,
Re: Response for manuscript ID: polymers-2144382 “Synergistic Flame Retardant Properties of Polyoxymethylene with Surface Modified Intumescent Flame Retardant and Calcium Carbonate”
Dear reviewer #2,
Thanks for providing us with this great opportunity to submit a revised version of our manuscript. We appreciate the detailed and constructive comments provided by the reviewers. We have carefully revised the manuscript by incorporating all the suggestions by the review panel.
We hope this revised manuscript has addressed your concerns, and look forward to hearing from you.
Sincerely,
The Authors
-------------------------------------------------------------------------------
Reply to Reviewer #2
Dear Reviewers,
Thank you very much for your time involved in reviewing the manuscript and your very encouraging comments on the merits.
Comments:
“187 Result of FTIR need some explanation about the reason why the functional group that was found to be similar or changed.
As well as the newly found spectrum brings what to the sample.
195 Similar with XRD. The different diffraction brings what to the sample. Need some explanation and reasons.
213 How the UL94 test could determine type of gas released?
217 How the compactness property could be related in flammability test?
There need to be some explanation about how the different additives added reacted during flammability test & LOI, due to difference in flammability results.
240 I think a citation is good to be here.
Most of the explanation, normally we add citation as we refer the similarity or the reason of the findings.
248 Need some explanation on this.
Overall
It is a very good and well-established article.
However it needs some proofreading at the early part (1.0, 2.0)
Well done.”
We also appreciate your clear and detailed feedback and hope that the explanation has fully addressed all of your concerns. In the remainder of this letter, we discuss each of your comments individually along with our corresponding responses.
To facilitate this discussion, we first retype your comments in italic font and then present our responses to the comments.
Comment 1:
187 Result of FTIR need some explanation about the reason why the functional group that was found to be similar or changed.
As well as the newly found spectrum brings what to the sample.
Response 1:
Thank you very much for your suggestion, which we have elaborated and explained in more detail in the FTIR characterization.
Comment 2:
195 Similar with XRD. The different diffraction brings what to the sample. Need some explanation and reasons.
Response 2:
Thank you very much for your advice, and we have also carried out more detailed elaboration and explanation in the XRD characterization.
Comment 3:
213 How the UL94 test could determine type of gas released?
Response 3:
Your prompt is very necessary, we have combed through the analysis of UL-94 and made a new discussion.
Comment 4:
217 How the compactness property could be related in flammability test?
There need to be some explanation about how the different additives added reacted during flammability test & LOI, due to difference in flammability results.
Response 4:
Thank you very much. In general, the higher the density of the carbon layer, the more effective it is to block the exchange of combustible substances inside the material and external heat sources. Therefore, the density of the carbon layer is related to the flame retardant performance of the material. Our description here is rather vague, so in the paper, we will explain and elaborate in more detail.
Comment 5:
240 I think a citation is good to be here.
Most of the explanation, normally we add citation as we refer the similarity or the reason of the findings.
Response 5:
Thanks for your prompt, we have included the citation here.
Comment 6:
248 Need some explanation on this.
Response 6:
Thank you, we have added more explanations to the manuscript.
We would like to take this opportunity to thank you for all your time involved and this great opportunity for us to improve the manuscript. We hope you will find this revised version satisfactory. Finally, we once again apologize for not being able to give you a valid response earlier.
Sincerely,
The Authors
-----End of Reply to Reviewer #2------

Reviewer 3 Report
Synergistic flame retardant effect of Ti-APP and CaCO3 on POM was studied in this paper, and perfect flame retardant performance was obtained. Thorough characterization was carried out. This paper is acceptable after some revisions:
In Introduction, the relative study on flame retardant of POM itself is too less.
Line 83-84, there shall be reference citation for the previous study.
Surface treatment of APP and CaCO3 is strongly influenced by the particle size. So in section 2.2, the particle size of APP and CaCO3 shall be given.
POM is quite easy to decompose during thermal processing, so carefully designed stabilizer package is necessary for POM. In Table 1, only 0.5% antioxidant is added, please make sure that no decomposition happened during extrusion and injection.
Figure 4 and Table 2 just show that after surface treatment, there are more organic component in modified APP and CaCO3. There is meaningless to discuss their thermal stability.
Table 3 shows that surface treatment of APP play a key function, and Ti-APP and CaCO3 have synergistic effect, while surface treatment of CaCO3 has little effect or even do a little harm?
In Figure 10, peak at 1630cm-1 is more likely to be attributed to absorbed water other than C=O, since 3431cm-1 is also sttributed to absorbed water.
Introduction of as much as 30% flame retardant often do harm to the mechanical properties, I suggest to add this data. The ideal result is expected to achieve good flame retardant performance at a little loss of mechanical properties. If not, it is possible to decrease the loading of flame retardant, since very good flame retardant performance has been achieved at loading of 30%.
There are some spelling mistakes in this paper, e.g., in the authors, in line 77.
Author Response
January 10, 2023
Prof. Fang,
Re: Response for manuscript ID: polymers-2144382 “Synergistic Flame Retardant Properties of Polyoxymethylene with Surface Modified Intumescent Flame Retardant and Calcium Carbonate”
Dear reviewer #3,
Thanks for providing us with this great opportunity to submit a revised version of our manuscript. We appreciate the detailed and constructive comments provided by the reviewers. We have carefully revised the manuscript by incorporating all the suggestions by the review panel.
We hope this revised manuscript has addressed your concerns, and look forward to hearing from you.
Sincerely,
The Authors
-------------------------------------------------------------------------------
Reply to Reviewer #3
Dear Reviewers,
Thank you very much for your time involved in reviewing the manuscript and your very encouraging comments on the merits.
Comments:
“In Introduction, the relative study on flame retardant of POM itself is too less.
Line 83-84, there shall be reference citation for the previous study.
Surface treatment of APP and CaCO3 is strongly influenced by the particle size. So in section 2.2, the particle size of APP and CaCO3 shall be given.
POM is quite easy to decompose during thermal processing, so carefully designed stabilizer package is necessary for POM. In Table 1, only 0.5% antioxidant is added, please make sure that no decomposition happened during extrusion and injection.
Figure 4 and Table 2 just show that after surface treatment, there are more organic component in modified APP and CaCO3. There is meaningless to discuss their thermal stability.
Table 3 shows that surface treatment of APP play a key function, and Ti-APP and CaCO3 have synergistic effect, while surface treatment of CaCO3 has little effect or even do a little harm?
In Figure 10, peak at 1630cm-1 is more likely to be attributed to absorbed water other than C=O, since 3431cm-1 is also sttributed to absorbed water.
Introduction of as much as 30% flame retardant often do harm to the mechanical properties, I suggest to add this data. The ideal result is expected to achieve good flame retardant performance at a little loss of mechanical properties. If not, it is possible to decrease the loading of flame retardant, since very good flame retardant performance has been achieved at loading of 30%.
There are some spelling mistakes in this paper, e.g., in the authors, in line 77.”
We also appreciate your clear and detailed feedback and hope that the explanation has fully addressed all of your concerns. In the remainder of this letter, we discuss each of your comments individually along with our corresponding responses.
To facilitate this discussion, we first retype your comments in italic font and then present our responses to the comments.
Comment 1:
In Introduction, the relative study on flame retardant of POM itself is too less.
Response 1:
Thank you very much. We have added some contents related to flame retardant polyformaldehyde in introduction. At present, there are very few research reports on flame retardant POM resin. Therefore, our laboratory is still working hard to solve the problem of flame retardant POM resin.
Comment 2:
Line 83-84, there shall be reference citation for the previous study.
Response 2:
Thank you very much. We have inserted the references in the appropriate places.
Comment 3:
Surface treatment of APP and CaCO3 is strongly influenced by the particle size. So in section 2.2, the particle size of APP and CaCO3 shall be given.
Response 3:
Your suggestion is very necessary. We have added the particle size and SEM image of CaCO3 in section of "2.1Materials".
Comment 4:
POM is quite easy to decompose during thermal processing, so carefully designed stabilizer package is necessary for POM. In Table 1, only 0.5% antioxidant is added, please make sure that no decomposition happened during extrusion and injection.
Response 4:
Thank you very much. Our experimental formula has been verified by long time and repeated experiments, which makes the possibility of decomposition of POM in the process of processing is relatively low, and no obvious decomposition phenomenon was observed in this experiment.
Comment 5:
Figure 4 and Table 2 just show that after surface treatment, there are more organic component in modified APP and CaCO3. There is meaningless to discuss their thermal stability.
Response 5:
Thank you very much. The main purpose of thermal analysis of APP and CaCO3 is to characterize their thermal stability before and after modification, as well as their flame retardant compatibility with the matrix resin.
Comment 6:
Table 3 shows that surface treatment of APP play a key function, and Ti-APP and CaCO3 have synergistic effect, while surface treatment of CaCO3 has little effect or even do a little harm.
Response 6:
Thank you. Indeed, as you mentioned, modification of CaCO3 does not seem to be conducive to flame retardant systems. We still keep it in the manuscript, mainly for comparison, to highlight that Ti-APP+CaCO3 is the optimal system.
Comment 7:
In Figure 10, peak at 1630cm-1 is more likely to be attributed to absorbed water other than C=O, since 3431cm-1 is also sttributed to absorbed water.
Response 7:
Thank you for the tip. The peak at 1630cm-1 may be water absorption, but according to the literature, it is also possible that C=O is present in the carbon layer, and the carbon layer has been well protected in the experiment.
Comment 8:
Introduction of as much as 30% flame retardant often do harm to the mechanical properties, I suggest to add this data. The ideal result is expected to achieve good flame retardant performance at a little loss of mechanical properties. If not, it is possible to decrease the loading of flame retardant, since very good flame retardant performance has been achieved at loading of 30%.
Response 8:
Thank you for your suggestion. The mechanical properties of POM/Ti-IFR and POM/Ti-IFR/CaCO3 are basically the same as those of unmodified APP, without obvious improvement. So, we only focus on flame retardant properties. The loading capacity and ratio of IFR were obtained through our previous exploration and orthogonal experiment, so they were not changed in this study.
Comment 9:
There are some spelling mistakes in this paper, e.g., in the authors, in line 77.
Response 9:
Thank you for your careful review of our manuscript. We have corrected the omissions and revised the manuscript again.
We would like to take this opportunity to thank you for all your time involved and this great opportunity for us to improve the manuscript. We hope you will find this revised version satisfactory. Finally, we once again apologize for not being able to give you a valid response earlier.
Sincerely,
The Authors
-----End of Reply to Reviewer #3------

Round 2
Reviewer 1 Report
Authors have carefully answered the reviewer's questions, and the current version can be accepted and published.